# Molecular Analysis in a Glioblastoma Cohort—Results of a Prospective Analysis

**DOI:** 10.3390/jpm12050685

**Published:** 2022-04-26

**Authors:** Liverana Lauretti, Tonia Cenci, Nicola Montano, Martina Offi, Martina Giordano, Valerio M. Caccavella, Antonella Mangraviti, Ludovico Agostini, Alessandro Olivi, Lucia Gabriele, Luigi M. Larocca, Roberto Pallini, Maurizio Martini, Quintino Giorgio D’Alessandris

**Affiliations:** 1Department of Neurosurgery, Fondazione Policlinico Universitario A. Gemelli IRCCS, Università Cattolca del S. Cuore, Largo A. Gemelli, 8, 00168 Rome, Italy; liverana.lauretti@unicatt.it (L.L.); nicola.montano@policlinicogemelli.it (N.M.); martinaoffi.mo@gmail.com (M.O.); marty-gio@live.it (M.G.); valeriom.caccavella@gmail.com (V.M.C.); antomangraviti@gmail.com (A.M.); ludoago16@gmail.com (L.A.); alessandro.olivi@policlinicogemelli.t (A.O.); quintinogiorgio.dalessandris@policlinicogemelli.it (Q.G.D.); 2Department of Pathology, Fondazione Policlinico Universitario A. Gemelli IRCCS, Università Cattolca del S. Cuore, Largo A. Gemelli, 8, 00168 Rome, Italy; tonia.cenci@gmail.com (T.C.); luigimaria.larocca@unicatt.it (L.M.L.); maurizio.martini@unicatt.it (M.M.); 3Department of Oncology and Molecular Medicine, Istituto Superiore di Sanità, Viale Regina Elena, 299, 00161 Rome, Italy; lucia.gabriele@iss.it

**Keywords:** EGFRvIII, glioblastoma, female, VEGF, trial population

## Abstract

The prognostic role of epidermal growth factor receptor variant III (EGFRvIII), a constitutively activated oncogenic receptor, in glioblastoma is controversial. We performed a prospective study enrolling 355 patients operated on for de novo glioblastoma at a large academic center. The molecular profile, including EGFRvIII status, MGMT promoter methylation, and VEGF expression, was assessed. Standard parameters (age, clinical status and extent of surgical resection) were confirmed to hold prognostic value. MGMT promoter methylation portended a slightly improved survival. In the whole series, confirming previous results, EGFRvIII was not associated with worsened prognosis. Interestingly, female sex was associated with a better outcome. Such findings are of interest for the design of future trials.

## 1. Introduction

This study analyzes a consecutive series of 355 patients operated upon for de novo glioblastoma (GBM). Age, post-operative neurological condition (Karnofsky performance status, KPS), and extent of surgical resection were confirmed to hold significant prognostic value. Interestingly, female sex was associated with a better outcome. The epidermal growth factor receptor (EGFR) variant III (EGFRvIII) is a constitutively activated mutated form of EGFR, which has been linked to increased proliferation and invasiveness of GBM cells in preclinical models [1]. Thus, it was traditionally deemed to carry a negative prognostic value. However, we previously showed the positive prognostic value of EGFRvIII expression in GBM patients, homogeneously treated with radiotherapy and concomitant and adjuvant temozolomide [2]. Since then, several studies have tried to assess the prognostic role of EGFRvIII in GBM, without definitive results [3]. In the 2021 WHO classification of central nervous system tumors, EGFR amplification, but not EGFRvIII expression, is a diagnostic criterion for GBM, IDH-wildtype, when histopathological criteria do not allow for definitive diagnosis [4]. However, the expression of EGFRvIII has been closely linked to EGFR gene amplification, since up to 50–60% of EGFR-amplified GBMs express EGFRvIII [5]. EGFRvIII has also been proposed as a therapeutic target, with some success [6,7].

The aim of the present paper was to assess the prognostic role of EGFRvIII in a large single-institution prospective cohort of de novo GBMs.

## 2. Materials and Methods

### 2.1. Patient Enrollment

We enrolled 355 consecutive patients operated upon for de novo GBM at the Department of Neurosurgery of Fondazione Policlinico Gemelli, Rome, between January 2012 and December 2017, in whom the expression of EGFRvIII was prospectively assessed. Patients operated upon in this timeframe for the recurrence of de novo GBM were also enrolled; however, clinical and molecular data referred to the first surgery. Patients for whom data from the first surgery were not available were excluded. Diagnosis was established using the criteria set forth in the 2007 WHO classification of Central Nervous System Tumors [8] for patients operated until 2016, when a new classification was set in place [9]. After 2016, only patients harboring GBM, IDH-wildtype were enrolled. All patients were treated as per the standard of care [10]. This study was approved by the Ethics Committee of Fondazione Policlinico Gemelli (study ID 1722).

### 2.2. Data Collection

The extent of surgical resection was judged based either on the surgeon’s impression or, where available, on postoperative MRI performed 24–72 h after surgery. Overall survival (OS) was defined as the time interval between tumor diagnosis and death from any cause.

### 2.3. Molecular Characterization

The molecular profile of the tumor was assessed as previously described [2,6,11]. Briefly, the proliferation index (Ki-67) and VEGF expression were assessed by immunohistochemistry. IDH status was assessed either by immunohistochemistry (anti-IDH1R132H antibody) or by mutational analysis of IDH1 and IDH2 genes [12]. O6-methylguanine DNA methyltransferase (MGMT) promoter methylation status was assessed using methylation-specific PCR, and EGFRvIII expression was assessed using RT-PCR.

### 2.4. Statistical Analysis

Survival curves were plotted using the Kaplan–Meier method and analyzed using the log-rank test. Multivariate analysis for survival was performed using Cox proportional hazards model. Comparison of categorical variables was performed using the Fisher exact test. All *p*-values were based on two-tailed tests, and differences were considered significant when *p* < 0.05. StatView ver5.0 was used (Sas Institute, Cary, NC, USA).

## 3. Results

### 3.1. Clinical Characteristics

Clinical features of this series are provided in Table 1. Most patients were male (62%), aged 65 years or younger (52.4%). Their postoperative KPS was >70 in 69.1% of cases. Gross-total resection was achieved in 68.7% of cases. In the whole cohort, the median OS was 13 months (13.5 months when excluding biopsy patients).

### 3.2. Molecular Features

The molecular characteristics of tumors are given in Table 2. IDH status was assessed in 52% of cases; all of these cases harbored IDH-wildtype GBM. EGFRvIII was positive in about half of the cases. MGMT promoter was methylated in 56% of cases. VEGF was hyper-expressed in the vast majority of patients.

### 3.3. Standard Prognosticators

Age, KPS and extent of resection are established prognostic factors in GBM, and this notion was confirmed in our series (Table 3 and Figure 1). Consistently with recent reports [13], female sex was associated with better prognosis (Table 3 and Figure 1). Of note, no significant differences between female and male patients were noticed in the proportion of patients aged < 65 years (50.7% vs. 53.0%, respectively; *p* = 0.7421; Fisher exact test) or in the proportion of gross-total resection cases (67.9% vs. 69.4%; *p* = 0.8131; Fisher exact test). Instead, the female cohort was enriched with cases with a KPS < 70 (40.6% vs. 24.7% in males; *p* = 0.0026; Fisher Exact Test), thus reinforcing the prognostic value of female sex per se.

### 3.4. Molecular Prognosticators

The well-known positive prognostic role of MGMT promoter methylation was also confirmed in our series, though with a low significance (median OS 13 months in methylated vs. 12.5 months in unmethylated patients, *p* = 0.0555, log-rank test; Table 3 and Figure 2). Instead, EGFRvIII expression was not prognostic when the whole series was analyzed (Figure 2 and Appendix A). As expected, the proliferative index and VEGF expression were not endowed with prognostic value.

Since most GBM trials enroll young patients with a good performance status, IDH-wildtype status, and unmethylated MGMT promoter [14,15], we investigated the prognostic role of EGFRvIII in this subgroup (*n* = 27). Intriguingly, we found that median OS was longer in EGFRvIII-positive than in EGFRvIII-negative patients (18.5 vs. 13.5 months), though the difference was not statistically significant (Figure 3).

In multivariate analysis, age, sex, KPS, and extent of resection had an independent prognostic role for OS, while MGMT had a minor role. EGFRvIII expression trended to be significantly associated with prognosis (Table 4).

## 4. Discussion

The present work aimed at assessing the impact of EGFRvIII expression on the outcome of patients with de novo GBM. In a previous study on 73 patients, we showed that, notwithstanding its role as oncogenic driver, EGFRvIII expression was not associated with a worsened prognosis [2]. In that series, EGFRvIII-positive patients had a superior median OS compared to EGFRvIII-negative patients (19 vs. 10.5 months). This evidence was explained by an increased sensitivity to temozolomide of EGFRvIII-positive tumors, an assumption that has also been confirmed in other papers [16]. Subsequent studies, however, were not able to unambiguously demonstrate the favorable or detrimental role of EGFRvIII expression in GBM [3]. Interestingly, a phase III trial enrolling only EGFRvIII-positive GBM patients to assess the effectiveness of an anti-EGFRvIII peptide vaccine (ACT IV) [7] reported very long OS both in the treatment and in the control arms. This study reported a 20-month median OS, a figure remarkably longer than those reported in the landmark temozolomide trial (14.6 months) [17] and in the more recent bevacizumab trials (15.7–16.7 months) [18,19]. Although the authors claimed that the better OS was due to the improvement of standards of care over time, one could speculate that the expression of EGFRvIII itself conferred prolonged survival.

Overall, EGFRvIII expression was not endowed with a significant prognostic role in our series. Looking for subgroups of patients in which EGFRvIII could have a prognostic role, we focused on the typical GBM trial population. Currently, GBM trials tend to enroll patients in good clinical conditions and with an unmethylated MGMT promoter, since in these cases it is considered ethically sound to omit temozolomide [14,15]. Interestingly, when we analyzed this subgroup of patients, we found that EGFRvIII conferred a remarkable, albeit not significant, survival advantage (Figure 3). Our 18.5-month median OS is quite similar to the 20-month median OS of the ACT IV trial [7]. This evidence needs confirmation in larger ad hoc series.

An important finding of the present study is the longer survival of the female GBM patients compared to the male ones. Traditionally, GBM was reported to occur more frequently in males, but no differences in prognosis between genders were demonstrated. It was only recently that such evidence emerged [13]. Since the immune tumor microenvironment is sex-specific [20], differences in immunosuppressive infiltrate between genders may help in explaining this issue [21]. However, other possible explanations, including a role of sex hormones in GBM oncogenesis [13], can be hypothesized and further research on this topic is warranted.

Strong points of the present work are the prospective design of the study and the number of enrolled patients. The prognostic value of standard prognosticators was here confirmed, assuring the generalizability of our results. As limitations, the single-institution design could weaken generalizability, although it ensured that patients were uniformly treated. Only the OS was intentionally recorded, since the progression-free survival suffers from the interpretation of response and progression.

## 5. Conclusions

To conclude, EGFRvIII expression does not hold prognostic value in unselected GBM patients; however, it could impact the prognosis of particular subgroups of cases, as trial-candidate patients. The improved outcome of female patients prompts mechanistic studies.

## Figures and Tables

**Figure 1 jpm-12-00685-f001:**
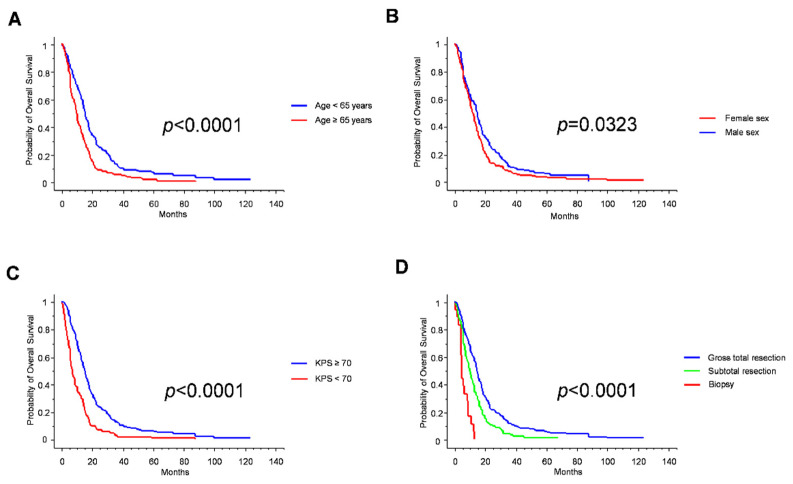
Kaplan–Meier survival curves for age (**A**), sex (**B**), Karnofsky performance status (**C**) and extent of resection (**D**).

**Figure 2 jpm-12-00685-f002:**
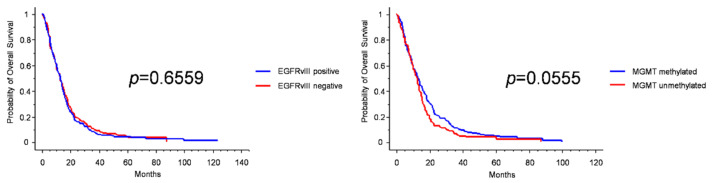
Kaplan–Meier survival curves for EGFRvIII expression (**left**), and MGMT promoter methylation (**right**).

**Figure 3 jpm-12-00685-f003:**
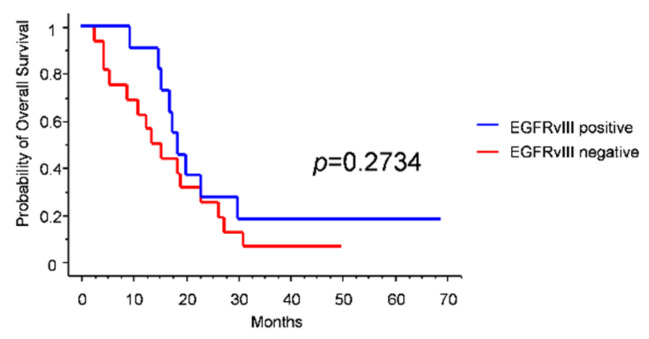
Kaplan–Meier survival curves for EGFRvIII expression in the trial-like subgroup.

**Table 1 jpm-12-00685-t001:** Clinical characteristics of enrolled patients.

Parameter	Result
*n*	355
Age (mean ± SD)	62.5 ± 10.6 years
Sex, M:F (%)	219:134 (62–38%)
Symptom duration (mean ± SD)	1.5 ± 1.6 months
Tumor diameter (mean ± SD)	4.7 ± 1.5 cm
Postoperative KPS (median, range)	70 (90–20)
Tumor location (%)	
frontal	109 (30.8%)
temporal	150 (42.4%)
parietal	54 (15.3%)
other	29 (8.2%)
multicentric	13 (3.7%)
Extent of resection	
GTR	244 (68.7%)
STR	93 (26.2%)
biopsy	18 (5.1%)
Median OS	13 months

GTR, gross-total resection; KPS, Karnofsky performance status; STR, subtotal resection.

**Table 2 jpm-12-00685-t002:** Molecular characteristics of patients.

Parameter	Result
EGFRvIII	
positive	184 (51.8%)
negative	171 (48.2%)
MGMT promoter	
methylated	198 (56.3%)
unmethylated	154 (43.8%)
VEGF	
hyperexpressed	235 (87%)
not hyperexpressed	35 (13%)
Proliferation index, median (range)	35% (4–70)

**Table 3 jpm-12-00685-t003:** Univariate analysis for survival.

Parameter	Median OS	*p*-Value
Whole series	13 months	NA
Age		<0.0001
≥65 years	15.5 months	
<65 years	10 months	
Sex		0.0323
Male	12 months	
Female	15 months	
KPS		<0.0001
≥70	15.5 months	
<70	7.5 months	
EOR		<0.0001
GTR	14.5 months	
STR	10 months	
biopsy	4.5 months	
EGFRvIII		0.6559
positive	13 months	
negative	13 months	
MGMT promoter		0.0555
methylated	13 months	
unmethylated	12.5 months	
Proliferative index		0.2804
≥30%	13 months	
<30%	13 months	
VEGF		0.8747
hyperexpressed	13 months	
not hyperexpressed	11 months	

EOR, extent of resection; KPS, Karnofsky performance status.

**Table 4 jpm-12-00685-t004:** Cox multivariate analysis for survival.

Parameter	Hazard Ratio	95% Confidence Interval	*p*-Value
Age	0.564	0.449–0.709	<0.0001
Sex	0.624	0.492–0.791	<0.0001
KPS	0.45	0.348–0.581	<0.0001
Extent of resection	1.970	1.543–2.514	<0.0001
EGFRvIII	0.813	0.650–1.016	0.0687
MGMT	0.941	0.748–1.184	0.6038

KPS, Karnofsky performance status.

## Data Availability

Source data are available from the corresponding author upon reasonable request.

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
