# Peer review of "Molecular Analysis in a Glioblastoma Cohort—Results of a Prospective Analysis"

_jpm, 2022, doi:10.3390/jpm12050685_

Round 1

Reviewer 1 Report

The authors describe a prospective cohort of glioblastoma cases in which they follow and assess for a set of factors and their association with overall survival. A strength of this study is the prospective nature. The concordance with factors known to associate with survival (such as gross total resection, KPS score, and age) is reassuring.

Comment #1: A major limitation is that half of the cases do not have assessment for IDH status. IDH is a known major predictor of outcomes in patients with tumors with histologic features of glioblastoma. IDH mutations are nearly mutually exclusive with EGFRvIII. Thus, When comparing EGFRvIII-mutated tumors to those without it, the cohort lacking EGFRvIII mutations is likely to be contaminated with some IDH-mutant cases. Therefore, it is not possible to truly assess prognostic implications in a cohort for which IDH status has not been addressed, as the EGFRvIII-wildtype cohort will be contaminated with IDH-mutant gliomas. Two possible ways to address: Ideally, if IDH sequencing can be performed on all tumors, a more definitive assessment for impact of EGFRvIII mutations could be performed. Given the author's suggestion that EGFRvIII might be suggestive of improved survival, this would be ideal, as removing IDH-mutant gliomas could possibly clarify this. If not possible, then a multivariate analysis in a cohort of patients with known IDH-wildtype status could be considered. It is difficult to draw conclusions from the very restrictive cohort used that had only 27 patients in it, and as such, the ability to draw conclusions on the impact of EGFRvIII mutations requires a larger robust cohort.

Comment #2. In the abstract, it is unclear what is meant by the statement "am ideal glioblastoma trial population." The data here does not appear to support EGFrvIII expression being associated with improved survival, and caution should be used in making such strong statements based on single survival curve with only 27 total patients for which the curves appear to  nearly intersect at around 23 months.

Comment #3: In the discussion, the authors note the increased survival for female patients, and suggest that this is due to differences in the immune microenvironment. While this is certainly a possible explanation, the reasons for the difference in survival remain poorly understood and other possible explanations remain, which should be discussed at least briefly here.

Author Response

Response to Comment #1: We agree with the Reviewer’s criticism. Though we carefully selected only de novo GBMs, it is not possible to rule out the possibility that some of the EGFRvIII-negative cases could be IDH-mutant. Therefore, we analyzed the prognostic role of EGFRvIII in the subgroup of confirmed IDH-wildtype cases: results are shown in Supplementary Figure S1 and described in Results section, page 5, lines 128-129. In the whole IDH-wildtype GBM series, EGFRvIII expression is not a prognostic marker, similarly to what is already shown in Figure 2. Instead, EGFRvIII positive/p53 positive cases had a significantly good prognosis (p=0.0176), reinforcing the trend evidenced in Figure 2. Regarding the small “trial-like” 27-patient cohort, we acknowledge that definitive conclusions cannot be drawn from such a small subgroup. We thus removed the word “remarkably” in the Results section (page 5, line 133) and, in the Discussion section, we added: “This evidence needs confirmation in larger ad hoc series” (page 6, lines 168-169).

Response to Comment #2: in the abstract (lines 25-26) we specified what we meant with the definition of ideal trial population (“young and fit patients harboring IDH-wildtype, MGMT unmethylated GBM”). To reduce the strength of the statement, in line 26 we changed the word “could” with the word “might”.

Response to Comment #3: thank you for this comment. Another point that has been raised in the literature is the oncogenic role of sex hormones. We have added a comment in the Discussion section, page 6, lines 181-183.

Please see also the revised Cover Letter.

Reviewer 2 Report

In the manuscript titled “Prognostic significance of clinicopathological features and EGFRvIII expression. Prospective analysis in a glioblastoma 3 cohort” Lauretti et al. investigate the prognostic role of a constitutively active oncogenic receptor, EGFRvIII, expression. The authors perform a prospective study in 355 patients operated for de novo GBM. The authors report that EGFRvIII expression is not associated with a worse prognosis. In the context of p53 positive tumors, EGFRvIII expression correlates with better overall survival. Additionally, as expected, the authors report that age, clinical status, and extent of resection hold good prognostic value, and MGMT promoter methylation correlated with marginally improved survival. Finally, the authors also report female sex correlates with better overall survival.

The finding that female sex correlates with better overall survival is surprising and potentially interesting. However, it is unclear if parameters such as age, clinical status, and extent of resection are similar between male and female groups. The authors should consider adding this information to their analysis. Overall, the study is well-executed, and the manuscript is easy to read. The study will be of interest to GBM investigators and will be an important data point for future meta-analyses exploring the role of EGFRvIII expression in GBM. I recommend publication of this manuscript in JPM.

Author Response

We thank the Reviewer for his/her positive comments to our work. We have compared baseline characteristics between female and male patients. We have found no differences in terms of age and extent of resection, whereas, surprisingly, female patient cohort had a higher percentage of poor KPS cases. This finding reinforces the independent positive prognostic role of female gender. We have added these data in the Results section, page 3 lines 102-108.

Please see also the revised Cover Letter

Round 2

Reviewer 1 Report

I thank the authors for their responses.

Comment #1: Supplemental Figure 1 is very convincing in demonstrating that there is no significant survival difference between EGFRvIII-mutant tumors and EGFR wildtype tumors in IDH-wildtype glioblastomas. This further brings into question the conclusions drawn for the small "optimal" cohort of GBM patients. I kindly suggest removing any reference to potential survival advantage in a very refined cohort from the abstract, as the number are too small to draw such a conclusion, and are not supported from the larger cohorts showing no survival difference.

Comment #2: I would interpret Supplemental Figure 1 panel B with great caution. p53 status was determined by immunohistochemistry, as it has great intraobserver variability and does not consistently correlate with mutation status. The papers referenced in the Methods for p53 IHC do not appear to describe any metrics for p53 interpretation or how cutoffs were determined, further limiting the utility of this metric. Therefore, I would recommend either sequencing for p53 in all cases where this is possible and performing the analysis in that cohort, or removing the p53 analysis due to this significant limitation of interpretation of IHC (Figure 3 and Supplemental Figure 1).

Author Response

Response to comment #1: According to your request, we removed the reference to the small “optimal” cohort of GBM patients from the abstract.

Response to comment #2: According to your request, we removed any reference to p53 analysis throughout the text, tables and figures.